# Reports on sexual violence published in an online Chinese newspaper: A new frame research

Yang Liu[1], Zhongzheng Fang[2]*

1 School of Literature and Journalism, Xihua University, Chengdu, Sichuan Province, China, 2 Faculty of Global Business Administration, Anyang University: Anyang City, Gyeonggi-do, South Korea

* fzz123760@anyang.ac.kr

## Abstract

The leading role of the media is very important in the new media era to build the reporting system and framework of sexual violence crimes, guide people's awareness and public opinion, and improve society's vigilance on sexual crimes. This study took People's Daily Online, a representative online media in China, as a research object to analyse the reporting of sexual crimes over the past 15 years. We conducted relevant searches for specific key-words set in the Python crawler and used IBM SPSS Statistics 19 software to analyse the frequency of relevant content. The results of the research show that, firstly, there have been significant changes in the number of news stories about sexual crimes. Second, the majority of sexual crime news stories are from mainland China. Third, the focus of the news stories and people is relatively concentrated on the perpetrators. Fourth, the People's Daily Online's coverage of sexual crimes focuses on blaming the perpetrators. Fifth, sexual crimes show that the framework is more episodic. This paper examines changes in the coverage of sexual crimes in China and captures how the media cover socially relevant issues, providing important insights for future social health, psychological awareness and diversion, and media policy.

## 1. Introduction

The original purpose of Tarana Burke's use of 'Me Too' in 2006 was to empower women, particularly young and vulnerable women, through empathy. In October 2017, Alyssa Milano used the phrase to highlight the seriousness of the issue of sexual harassment and assault, showing how many people have experienced it first-hand. The actress and others launched the 'Me Too' movement, which has sparked a wider conversation about sexual harassment and assault outside of Hollywood in political, academic and cultural spheres. In 2022, Netflix screened a documentary about Room N, the sexual crime in South Korea that shook the world, with a thought-provoking sentence at the end: "The crime of imitating Room N continues to happen around the world." Sexual crimes are also common in China and cause heated discussions in society. Chinese society is no longer shying away from sexual crimes as it used to be. However, there are still many problems in the reporting of sexual crimes, so it is very

**Data Availability Statement:** Data availability The raw data supporting the conclusions of this article are available from the authors without undue reservation to any qualified researcher. The datasets analysed during the current study are

available in the Figshare repository: https://doi.org/10.6084/m9.figshare.24559792.v1.

**Funding:** The author(s) received no specific funding for this work.

**Competing interests:** The authors have declared that no competing interests exist.

important to find the relevant reported cases, summarise the reports, and present the reported contents objectively and fairly [1].

Most people do not directly experience social focus issues or events, but indirectly through the media. In news reports, sexual crime news often becomes the focus of social attention, mostly because of the sex of the persons in the crime incident, the relationship between the persons in the incident, the combination of crime and sex, etc., which can stimulate readers' curiosity. Crime is not only newsworthy, it affects everyone [2]. Similarly, Ian Marsh and Gaynor Melville's [3] book Crime, Justice and the Media contains a comprehensive analysis of crime coverage since 1945 and provides insights into the relationship between the media, public perceptions and crime reporting. Thus, the positive significance of crime news to society is obvious. It is not limited to letting the public know about the existence of crime cases and strengthening the social early warning function, but to embody legal justice and guarantee social justice.

The focus of each era can be analyzed and inferred by understanding and mastering what are important issues and values in a specific society. The media constitute news stories in specific ways according to the social order of the country and the interests of national stability [4]. Because of this, the news frame is widely used in news reports, and it can emphasize and highlight the thoughts and issues that want to be conveyed to the audience when selecting a variety of topics, information sources, and reporting techniques. The text of the news report shows the editors' different values and orientations and reflects their understanding and viewpoints, through which the reporter's intention and purpose of the report can also be understood.

The research on the news frame is still in its initial stage in China as a whole, which belongs to a relatively new research field and needs to be actively explored [5]. Therefore, it is necessary to make an in-depth study by means of mixed analysis in combination with China's social characteristics, media characteristics, and audience characteristics while understanding the theories of Western countries [6]. China's media circle should refer to the research at home and abroad to broaden the theoretical field of vision, especially exploring the news organization's intention of text production through frame analysis, which is of profound significance.

The purpose of this study is to provide an in-depth analysis of the coverage of sex crimes in recent years in China's main online media, the People's Daily Online. The study examines various aspects of this coverage, including the volume of sex crime news, the nature of the content, the areas of focus, the attribution of responsibility and the framing techniques used. This in-depth investigation aims to shed light on the Chinese media's attitudes and approaches to reporting sex crimes, as well as the impact of such reporting on societal attitudes and values.

In order to better understand these dynamics, this study asks several key research questions:

1. What are the characteristics and trends in reporting of sexual offences on the People's Daily Online?

2. What are the changes in the locations of sex crimes reported on People's Daily Online?

3. What are the trends in the form of sexual crime reporting on People's Daily Online?

4. What are the trends in the news sources of People's Daily Online's sex crime reports?

5. How often does People's Daily Online's report on celebrity sex crimes?

6. Is there a difference in media attention between perpetrators and victims in the People's Daily Online sex crime stories?

7. How does the People's Daily Online allocate responsibility between perpetrators and victims?

8. What is the relationship between thematic and situational reporting in People's Daily Online sex crime stories?

In the conclusion and discussion section, this study aims to highlight the important role that media coverage plays in raising social awareness of sexual crimes, shaping public ideology and upholding social justice. By providing a comprehensive and in-depth examination of the coverage of sexual crimes in China, this study offers valuable insights and recommendations that are relevant to future considerations in terms of social health, psychological awareness, media communication and policy development.

## 2. Conceptual definition and related literature

### 2.1 Conceptual definition

News coverage is shaped by the interaction of several complex factors that determine its structure and impact. Among these, the volume of coverage is the main indicator of the importance of news. The frequency and volume of coverage tend to indicate the value and urgency of an issue for the public. The quantity of news reports: the intrinsic importance of news is based on its quantity or frequency, linking the volume of coverage to its value [2]. The geographical context, especially the location of the crime, is also crucial. It provides a spatial dimension to the story, giving the audience a sense of proximity and relevance. Place of crime: the geographical area in which the criminal activity is reported, providing a spatial context for the crime. This is complemented by the form of reporting, which in today's multimedia journalism is varied, ranging from traditional written articles to digital and interactive formats. This diversity of presentation caters to the preferences of different audiences and influences the way news is consumed and interpreted. Report forms: The different forms of news reporting are indicative of the multimedia nature of modern journalism.

In news reporting, the source of information plays a crucial role in shaping the audience's perception. The information source of a report: As noted by Young-Jun Son and Juhyun Hong [7], the source of information contributes significantly to the suggestive effect. The origin, or primary source, from which information about a crime is derived influences the credibility and perspective of the report. The involvement of celebrities in news stories adds further complexity, as their social status can disproportionately influence the value of the news, often overshadowing other relevant aspects of the report. Whether the concerned is a celebrity or not: This aspect emphasises the impact of public figures on news value by highlighting their social recognition or status in news stories. In sexual crime reporting, the focus of media attention shifts between the perpetrator and the victim, influencing public opinion and sometimes leading to a distorted understanding of events. Media attention: Media coverage of sexual crime often focuses on 'who' is at the centre of the story—the perpetrator or the victim. In order to measure media attention, the analysis divides news topics and personal information publicity into categories such as "media attention to topics", "media attention to revealing people's information" and "emphasis on people-related content". The attribution of responsibility in crime reporting represents a delicate balance between the portrayal of the roles of perpetrator and victim, which can influence public discourse about such events. Attribution of case responsibilities: The attribution of responsibility in these reports is divided into frameworks that focus on either the perpetrator or the victim, analysing 'who is more responsible' in the context of sexual crime reports.

The thematic and episodic frames used in crafting news stories represent another dimension of influence. Thematic framing provides a broader narrative context, offering depth and background, whereas episodic framing hones in on specific events or instances, giving a more focused, immediate picture of the situation. The choice between these framing methods can significantly alter the audience's understanding and interpretation of the news, emphasizing the importance of thoughtful and balanced reporting. Thematic frame and episodic frame: The approach or perspective from which a news story is crafted, suggesting the broader narrative or the specific event it emphasizes. Together, these elements underscore the multifaceted nature of news reporting, highlighting the need for critical engagement and ethical considerations in both journalism and audience reception.

## 2.2 Related literature

According to previous research, 84.6% of women get information about sexual crimes through TV or newspaper reports. The formation of the media's attitudes and ideas toward the audience has had a significant impact [8], which requires a close understanding of events to capture the reader's interest and attention. Thus, the media will dramatically organize events and describe them in detail [9]. The news of sexual crimes is of high value because it contains elements that can attract readers' attention. Regardless of the combination of violence and sex, or the age of the victim of a sexual offence, it receives a high level of media attention [10].

Foreign research on media and sexual crimes has been carried out extensively. The first part is about the problems in the reports of sexual crimes. Scholars [11] analyzed the media reporting practices and the structural characteristics of media companies in South Korea on the coverage of the "Me Too Campaign", and concluded that various gender discrimination issues were exposed from the perspectives of the publishing, interview and reporting process, discussion process, and professional reporters reporting on women's focus issues within the enterprises. In addition, in the process of relevant reports, the competition of stop-press news in the news circulation environment centered on the portal website, the lack of understanding of the reporting policy of sexual crimes, and the contradictions in the myth of objectivist journalism have also been exposed. Secondly, the researcher distinguishes the identity of the perpetrator and the victim in the process of analyzing and reporting the differences between them. The study found [12] that although the victim-centered frame well represented their pain and trauma, the voice of the victims is reproduced through the radio, which not only fails to clarify the social structural problems that induce sexual crimes but triggers social debate. The media tried to convey the voice of the victims out of the sight of the perpetrators or through the practice of relying on investigative authorities. The problem, however, is that the structure of the press is still weak on the causes of victims of sexual crimes and how to deal with them [13].

To sum up, several issues need attention in the extensive research on media and sexual crimes. First, the analysis of sexual crime reporting requires a fundamental, structural cause analysis and a statistically significant basis for research that can generalize the results. In particular, the research on the reports on the perpetrators and victims and media attention is of great significance [2]. Second, communication studies should pay sustained attention to the construction of perpetrator and victim frames in order to construct more appropriate interpretive frameworks through the analysis and observation of actual media texts [14].

After examining the global perspectives on media reporting of sexual crimes, including the framing of narratives for perpetrators and victims as well as the subtle challenges faced by different countries, we now see that with the liberalization of the media in China, new challenges have also emerged in the reporting of sexual crimes. The media market in China is more liberal and personalized than before, and many events are revealed by the public before the media

intervenes and are made public in the form of news. However, there appeared a variety of anomie behaviors in related reports after the news was published, and people paid close attention to such issues as information disclosure, chat history, call recording, insufficient interviews in reported news, and biased reports. The anomie of sexual crime-related reports in China has occurred frequently in the past, so it is a problem that China media must pay attention to at present to reflect on the media standardization in sexual crime case reports [15].

The normative discussion on the news diffusion process of sexual crimes in China is far later than that in other countries, and there is a lack of scientific and convincing research in academic circles [16]. According to preliminary field studies, there are the following problems in studying sexual crime reports in China: First, most of the research on sexual crime in China has been conducted in law, education, medicine, sociology, psychology, etc., which is insufficient in the selection of online media to systematically construct sexual crime reports from the perspective of frame theory [17]. Secondly, there is little in-depth research on the types of sexual crimes in China. In most cases, it only raises questions, discusses the current situation and puts forward corresponding countermeasures through sexual crime news reports, almost with no empirical research on the causes and concrete development of the problems [18]. Although significant academic achievements have been made in the research direction, the research scope is wide and vague. The research object is biased toward children, teenagers, celebrities, and other specific groups. The research method is relatively simple, and it always descriptively asks several questions, or abstractly selects the questions with the same content to study, without systematic and rigorous explanation for the problems in the production of crime reports [5]. Most of the research results only focus on the management of state organs, and the research and analysis of some attempted sexual crime reports are too subjective and unconvincing. Third, the research process of news reports on crime types lacks scientific integrity. The research on sexual crime news mostly focuses on specific news at present, without making statistical analyses on the temporal changes of sexual crime news. In addition, it is insufficient to statistically analyze and process the number of news reports related to sexual crimes with correct data, which does not attach importance to the application of research methodology and lacks necessary scientific and empirical research.

The focus should be on the statistical analysis of the history of sexual crime news reports based on the comprehensive narration of the types of sexual crime news documents. The initial form, characteristics, changes, and progress are also very important in news reports. In this study, the characteristics of China's social-related reports will be tracked and analyzed based on journalism theory, and the temporal changes of reporting forms adapted to the changes of the times will be studied. In particular, the problems in the production of crime reports should be systematically and strictly explained, which is not only to establish the norms and ethics of media coverage of sexual crimes but also to carry out relevant research. At the same time, the media need to conduct scientific and systematic research from the perspective of communication on how to construct a reporting system and frame for sexual crimes in the new media era, to guide national understanding and public opinion, and to enhance social vigilance for sexual crimes.

Of course, the approach to reporting on such sensitive issues is not without its obstacles, and it is particularly important to recognise the impact of media strategies and the balance between sensationalism and responsibility in reporting on sexual offences. Coverage of sex crimes is expected to lead to social concern, increased public awareness and vigilance, and the development of relevant protective measures. However, the media will use provocative language to amplify events and stimulate public curiosity and voyeurism in order to gain public attention [19]. Maxwell McCombs and Sebastián Valenzuela [20], in Setting the Agenda: Mass Media and Public Opinion, revisit and extend Lippmann's [21] ideas to emphasise the key role of the news media in constructing our perception of reality. The media give their collected

event information logic in a variety of ways, including emphasizing any behavior or person, so the scale and frequency of media reporting events are very important in the issue-setting process. The reported quantity emphasizes the frequency and quantity of event reports from the perspective of media reports, while the spreading power of hot spots means how much influence hot spots have on society from the point of view of news value [22]. Therefore, it is very important to analyze the media content, because the media content analysis can not only infer those closed and invisible phenomena but also analyze and infer the audience, organization, and cultural background of specific content.

When reporting sexual crimes, the media not only reflects the pure purpose effect of attracting the attention of the audience by focusing on the news value from the perspective of the economic market but also includes the structural characteristics of the media environment, the comprehensive effects of journalists' personal feelings, the organizational culture of media companies, the competition between media companies, the production and distribution environment of digital news [8, 11]. Nevertheless, the role of the media in crime reporting is to provide relevant information about the incident, arouse the vigilance of the audience, and put forward solutions to prevent other losses. In particular, the report on crimes of sexual violence should discuss whether the media has put forward adequate solutions with a balanced view and whether the media has fulfilled its original role.

In addition, in a modern society in which a great deal of information is generated, it is very important to be able to understand, to judge and to grasp the information. Classifying, composing, and explaining the content through the frame can make people better understand the real world [23, 24]. The description of news reports not only reflects the simple individuals, media organizations, and phenomenal society but also reflects the social values, the reconstruction of specific realistic society, and other issues. The media not only have a cognitive impact on the public based on the number of reports but also have an important role in reporting those responsible for sexual crimes. "Attribution of the incident" means who is responsible for the incident, so it should be dealt with in an important way to clarify the responsibility of sexual crimes [25]. The study found that [26] media reporting events using a specific responsibility frame has the effect of inducing public awareness of the causal relationship between social focus issues. Therefore, it is very important and necessary to understand the reporting tendency of the offender and the victim as the center of sexual crime reporting.

Iyengar put forward the "thematic frame" and "episodic frame" in the form of TV news [27]. The study found that the vast majority of rape-type news reports used episodic statements. The formal descriptions are not just textual differences but also have an impact on the public's understanding of news and responsibility attribution. News reports play a role in providing contextual clues when readers judge the focus and responsibility of events. The audience pursues a simple understanding of the news content, usually relying on easy-to-imagine things to understand the attribution, and seldom considering the factors in an all-around way [28]. Therefore, it is necessary to study whether the news is carried out with thematic statements or anecdotal statements.

Through the discussion of the research, this paper deeply narrates the news statement theory, summarizes the attribution theory which serves as the theoretical basis of responsibility attribution, and then focuses on the responsibility attribution to describe the frame theory.

## 3. Materials and methods

### 3.1 Research design

In this study we explored in depth the issue of reporting of sexual offences over the 15 years since the People's Daily Online was established. The data collection, collation, clean-up, and

import phase took place from August 2021 to May 2022, and all authors were involved throughout the process. This article does not contain any studies with human participants or animals performed by any of the authors. The research process was carried out in four stages. First, in the data import phase, we used the Python programming language to run a web crawler (crawler) to automate the collection of sex crime-related news reports from the People's Daily Online (http://search.people.cn). Python is known for its powerful libraries and frameworks such as Beautiful Soup and Scrapy, which are widely used for web crawler development, and these tools can efficiently parse and extract web content.

Specifically, we used the Scrapy framework, written in Python, to automate the extraction of information from the People's Daily Online news archive. First, a Scrapy project was created and the necessary parameters set, and then the Item class was defined to explicitly specify the data fields to be extracted from the web page, such as the publication date, news content and link. Next, a specialised spider was written to locate the People's Daily Online news archive and, by setting the parameters to 'sexual crime', 'rape', 'sexual violence' and 'sexual assault', it was used to locate the news archive and automatically search for news articles related to these terms. The data was processed and stored using Scrapy's pipeline system, and all the data collected was organised in Excel. The second stage of data cleaning involved the precise screening of all the collected reports to ensure that only those directly related to sexual offences were included. In the third stage, I sampled the collected reports using a systematic sampling method. Finally, in the stage of statistical analysis and interpretation of information, IBM SPSS Statistics 19 software was used to perform frequency analysis of relevant content.

The time span of this study is from 1 January 1997, the date of the establishment of the People's Daily Online, to 31 December 2019, the date of its establishment. However, the actual time period analysed was determined based on the availability of relevant data, i.e. from 10 October 2004 for "sexual offences", "rape", "sexual violence" and "sexual assault", and from 10 October 2004 for "sexual offences", "rape", "sexual violence" and "sexual assault" to 31 December 2019 for "sexual offences", "rape", "sexual violence" and "sexual assault". "Sexual assault" from 10 October 2004 to 31 December 2019, when the keywords "sexual offence", "rape", "sexual violence" and "sexual assault" first appear in reports. This timeframe provides a comprehensive view of the evolution of coverage of sexual offences over time, capturing dynamic changes and trends in media coverage.

## 3.2 Material selection

As shown in Table 1, a keyword search was conducted using Python and a total of 80,385 stories were captured. Based on the results, 3637 stories related to "sexual crimes", 52,435 stories related to "rape", 3228 stories related to "sexual violence", 21085 stories related to "sexual assault", a total of 80385 stories. "Sexual assault" stories 21085, total 80385 stories. First, all the

**Table 1. Data collection and analysis process for sexual crime news study.**

| Stage | Description | Total Number |
|---|---|---|
| Initial Data Capture | News articles captured using Python | 80,385 |
| Data Cleaning | Articles screened based on definitions and study requirements | 13,897 |
| Systematic Sampling of Sexual Crime News | 5% of every 20 articles were sampled | 694 |
| Articles Before June 23, 2012 | Number of news pieces that had lapsed | 87 |
| Available Texts from Lapsed News | Texts that could be submitted from the lapsed news | 7 |
| Final Effective Analysis News | Total news pieces considered for effective analysis | 600 |

news items related to the four keywords were sorted in chronological order. Then the samples were screened according to the following requirements:

1) delete the news articles that repeated each other.

2) Delete "comfort women", "World War II" and historical content.

3) Delete news that is not related to sex crimes, such as "OO sex crimes" (e.g., violent sex crimes, group sex crimes, etc.).

4) Domestic violence, resource-based sex trafficking and other related issues are not included in this study.

5) Reporting of people with previous convictions for sexual offences for committing other crimes.

6) Misreporting (e.g. direct statements in the news, non-existent cases, etc.).

7) Cases where the same topic or news title is different but has exactly the same content on the same day.

8) Exclusion of content not related to sexual offences (e.g. film introductions, novel recommendations), etc.

Based on previous research, the research sample selected for this study is the series of news about sexual crimes, i.e., the occurrence of the case, the initiation of the police investigation, the arrest, the detention until the indictment, the indictment until the verdict of the trial, and the reporting and investigation of the case by the investigating body before the initiation of the investigation [29]. In addition, the study sample included news of cases of forced prostitution, news of clandestine filming, and published news of sexual violence, as well as comprehensive reports and comments on sexual violence. The data were cleaned according to the definitions and samples that did not meet the requirements of the study were excluded. In the end, 13,897 samples were collected.

Table 2 presents the variables used to study the coverage of sexual offences and provides a systematic overview of the key elements used to analyse the coverage of sexual offences on

**Table 2. Research variables in sexual crime news reporting.**

| Variable Category | Description | Details/ References |
|---|---|---|
| 1. Quantity of News Reports | Analysis of the quantity of sexual crime reports over time, as reported by People's Daily Online. | - |
| 2. Location of Cases | Location is divided into six zones: Chinese mainland, Hong Kong/ Macau/Taiwan, South Korea/Japan, other Asian countries, the US/ Europe/Australia, other countries (Africa, Latin America, etc.). | - |
| 3. Forms of News Reports | which includes traditional forms like text and image, as well as more modern forms prevalent in network media such as video, video and text, and video/picture/article combinations. | - |
| 4. Information Sources | Categorized into government agencies, NGOs, experts, involved parties, witnesses, and others. | - |
| 5. Inclusion of Celebrities | This aspect examines whether news reports feature entertainers, celebrities, or ordinary people, based on the social stratum of the persons in the news, as defined in prior research. | [2] [7] |
| 6. Media Attention on Perpetrators/ Victims | Examination of focus on perpetrators/victims, including topic emphasis, personal information disclosure, and related content. | [7] |
| 7. Attribution of Responsibility | This aspect evaluates how responsibility is attributed between the perpetrator and the victim in the reports. It uses a three-point scale to measure the extent to which the reports mention or emphasize the responsibility of either party. | [7] |
| 8. Thematic/Episodic Frame | Differentiation between thematic and episodic frames in news reports, including a combination of both, to understand reporting strategies. | [27] |

People's Daily Online. The table includes eight different categories, Quantity of News Reports, Location of Cases, Forms of News Reports, Information Sources, Inclusion of Celebrities, Media Attention on Perpetrators/ Victims, Attribution of Responsibility, Thematic/Episodic Frame, with precise descriptions for each category.

### 3.3 Data analysis

A systematic sampling method was used to select 5% of 20 articles at regular intervals, resulting in a total of 694 news articles on sexual violence. By 23 June 2012, 87 news articles in the test sample had expired and only 7 articles were available for submission. Thus, the final number of valid news articles for analysis was 600, with 5% of the sample coded by the coder and measured for intercoder reliability testing. To ensure the reliability verification of the analysis items, the reliability verification was carried out with the help of Professor Li Deshun, associate professor of communication at Huaiyin Normal University in China. The encoding was tested based on approximately 5% of the extracted 600 articles and a final random sampling of 30 articles.

To analyze the content objectively and ensure the reliability between codes, in this study, the Holsti method [30] commonly used in the content analysis was used to test the reliability of materials by percentage for mutual consistency. The inherent consistent reliability of the sequence problems was tested by the Cronbach α method.

The results showed that the reliability of each type of nominal problem was above 0.9, which verified the content reliability of the data used in this study. The sequence question was Cronbach's alpha = 0.658, which validated the reliability of the content of the data used in this study.

## 4. Research results

(1) Quantity of news reports about sexual crimes

In this study, the number of sexual crimes reported by the People's Daily Online was specifically calculated through data statistics, and the difference in reported quantity was analyzed based on time.

Table 3 shows the number of news related to sexual crimes each year. From 2004 to 2019, there were a total of 13,897 pieces of news on sexual crimes. On the whole, there were relatively few related reports from 2004 to 2010, and the first news of sexual crimes appeared on December 3, 2004. As of April 21, 2009, a total of 59 news articles were all related to Japan, with a slight increase in 2008 due to the occurrence of juvenile sexual crimes by the US military in Japan, accounting for 18 out of 34 news articles. From 2004 to 2010, the total number of news reports related to all sexual crimes was 71 (0.5%). In the early days, most of the news retrieval results about sexual violence articles in China were concentrated in Japan, especially the reports about comfort women, the Second World War, and other historical events. From 2011 to 2015, the number of relevant news articles exceeded 1,000 each year, especially in 2011, 2012, 2013, and 2014, which was significantly different from the number of original full-size specimens, because irrelevant news in the data editing process and the situation of repeating the same news in one day were excluded. Since then, from 2016 to 2019, news related to sexual crimes has once again shown a decreasing trend year by year.

(2) The place where the cases reported by the news about sexual crimes occurred

According to the specific actual situation, the locations of reported cases were divided into "Chinese mainland", "Hong Kong, China;Macau, China; and Taiwan, Province of China", "South Korea and Japan", "other Asian countries", "the United States, Europe and Australia", and "other countries (Africa, Latin America, etc.)", a total of six to be distinguished.

**Table 3. Number of news reports about sexual crimes in People's Daily Online.**

| Years | Number of news reports |
|---|---|
| 2004 | 2 (0.0%) |
| 2005 | 9 (0.1%) |
| 2006 | 3 (0.0%) |
| 2007 | 10 (0.1%) |
| 2008 | 34 (0.2%) |
| 2009 | 7 (0.1%) |
| 2010 | 6 (0.0%) |
| 2011 | 1,200 (8.6%) |
| 2012 | 2,423 (17.4%) |
| 2013 | 5,027 (36.2%) |
| 2014 | 2,745 (19.8%) |
| 2015 | 1,186 (8.5%) |
| 2016 | 463 (3.3%) |
| 2017 | 285 (2.1%) |
| 2018 | 231 (1.7%) |
| 2019 | 266 (1.9%) |
| Total | 13,897 (100.0%) |

Note: (unit: piece, (%))

According to Table 4, the results of frequency analysis on the places where sexual crimes are reported by People's Daily Online show that "Chinese mainland" had the largest number of reported sexual crimes with 242 cases (40.3%), followed by "other Asian countries" with 94 cases (15.7%). Although the number of news items in Chinese mainland was high on the whole, news from other regions as a whole accounted for 60% (358 pieces). That is to say, news from other regions combined accounted for more than half. Thus, the People's Daily Online reported more cases of sexual crimes in other regions than Chinese mainland.

(3) Forms of news reports related to sexual crimes

To understand the characteristics of sexual crime reports, three items including "text", and "text and image" were proposed. In addition, considering the characteristics of network media, five items including "video", "video and text", and "video, picture and article" were added.

According to Table 5, the results of frequency analysis on the forms of news reports about sexual crimes in People's Daily Online show that the forms of news reports about sexual crimes

**Table 4. Analysis results of places where cases of sexual crimes are reported by People's Daily Online.**

| Place of crime | Frequency | Percentage (%) |
|---|---|---|
| 1) Chinese mainland | 242 | 40.3 |
| 2) Hong Kong, Macau and Taiwan, China | 86 | 14.3 |
| 3) South Korea and Japan | 68 | 11.3 |
| 4) Other Asian countries | 94 | 15.7 |
| 5) The United States, Europe and Australia | 84 | 14.0 |
| 6) Other countries (Africa, Latin America, etc.) | 26 | 4.3 |
| Total | 600 | 100.0 |

Note: (unit: piece, (%))

**Table 5. Results of the analysis on the report form of the People's Daily Online news on sexual crimes.**

| Report form | Frequency | Percentage (%) |
|---|---|---|
| 1) Text | 421 | 70.2 |
| 2) text and image | 148 | 24.7 |
| 3) Video | 26 | 4.3 |
| 4) Video and text | 4 | 0.7 |
| 5) video, picture and article | 1 | 0.2 |
| Total | 600 | 100.0 |

Note: (unit: piece, (%))

are mainly "text". The forms of sexual crime news reports in People's Daily Online are more traditional statements.

(4) Information sources of news about sexual crimes

The information sources herein were analyzed in seven categories, namely, government agencies and their employees, non-governmental social organizations, experts and scholars, parties and their families, witnesses and relevant information providers on the spot, and unclear information sources and others.

According to Table 6, the results of frequency analysis of news sources of sexual crimes reported by People's Daily Online show that 395 (51%) news sources related to sexual crimes were "government agencies and their employees". On the whole, more than half of the news related to sexual crimes seek authoritative and reliable information sources such as "government agencies" and "experts and scholars". It thus can be judged that the sources of sexual crime-related news in China are generally obtained directly from the government and other highly trusted channels and the parties concerned.

(5) Is there a celebrity in the news reports about sexual crimes

According to the social stratum of the persons in the news in previous research [2]; [7], they were divided into "entertainers", "celebrities" and "ordinary people".

According to Table 7, the results of the frequency analysis on whether the persons related to sexual crimes reported by People's Daily Online are celebrities show that the persons related to sexual crimes reported by People's Daily Online are at most 500 "ordinary people" (83.3%), followed by 67 "entertainers" (11.2%). Thus, in the reports of People's Daily Online on sexual crimes, the related persons are mainly ordinary people, not celebrities or entertainers.

(6) Media attention on perpetrators/victims of sexual crimes-related news reports

**Table 6. Results of analysis on information sources of news on sexual crimes in People's Daily Online.**

| Information source | Frequency | Percentage (%) |
|---|---|---|
| 1) Government agencies and their employees | 395 | 51.0 |
| 2) Non-governmental social organizations | 44 | 5.7 |
| 3) Experts and scholars | 83 | 10.7 |
| 4) Parties and their families | 108 | 13.9 |
| 5) Witnesses and relevant information providers on the spot | 47 | 6.1 |
| 6) Unclear information sources | 54 | 7.0 |
| 7) Others | 44 | 5.7 |
| Total | 775 | 100.0 |

Note: (unit: piece, (%))

**Table 7. Analysis results of whether the relevant persons in the news reported by People's Daily Online about sexual crime are celebrities.**

| Celebrity or not | Frequency | Percentage (%) |
|---|---|---|
| 1) Entertainers | 67 | 11.2 |
| 2) Celebrity | 33 | 5.5 |
| 3) Ordinary people | 500 | 83.3 |
| Total | 600 | 100.0 |

Note: (unit: piece, (%))

In this study, the media attention of the perpetrators/victims of sexual crimes reported by People's Daily Online was analyzed from three dimensions, that is, the emphasis on news topics, the disclosure of personal information and the related content of persons, to analyze the media emphasis of all parts of the news.

To understand the centrality of victims or perpetrators in sexual crime reporting, the topics of the report were divided into "victim-centered", "neutral" and "perpetrator-centered" [7]. However, there might be some unclear topics in this study, so "other" was added.

The disclosure of personal information was also broken down according to the previous research which was set as "name", "age", "face", "address", "employer", "exposure of people around" and "exposure of private life". The measurement of the personal information index was based on the method of previous research [7]. To measure the personal information of the perpetrator and the victim, a 3-point scale was used, with 1 point for "not mentioned at all"; 2 points for "mentioning to some extent" and 3 points for "specifically mentioned". For example, if the name and age were made public, it was classified as "specifically mentioned"; if only the name was made public, it was classified as "mentioned to a certain extent"; if a pseudonym was used or the name was not mentioned, it was classified as "not mentioned at all" to calculate the sum of these items to measure the personal information disclosure index. The larger the value of the personal information index is, the more detailed and specific reports would be made on the personal information of the perpetrator or the victim.

Finally, the emphasis of person-related content refers to the reporting of the relevant situations and events in the form of personal interviews, with the victims or the perpetrators as the main body. Similarly, it was also divided into "victim-centered", "neutrality" and "perpetrator-centered".

Table 8 shows that 44.5% (267 articles) of the media coverage of sexual crime focused on 'perpetrators', while 21.3% (128 articles) focused on 'victims'. There is a difference of about 2 times between the news topic centred on the perpetrator and the news topic centred on the victim. The results of the analysis show that journalists pay more attention to perpetrator-related topics and use them to attract readers when editing news.

**Table 8. Results of the analysis on the topics of news reports on sexual crimes in People's Daily Online.**

| News topic | Frequency | Percentage (%) |
|---|---|---|
| 1) Neutrality | 74 | 12.3 |
| 2) Victim-centered | 128 | 21.3 |
| 3) Perpetrator-centered | 267 | 44.5 |
| 4) Other | 131 | 21.8 |
| Total | 600 | 100.0 |

Note: (unit: piece, (%))

Table 9 shows the degree of disclosure of personal information of perpetrators/victims of sexual crime reports that were analysed. As a result, the degree of disclosure of personal information of perpetrators was 9.9, and the degree of disclosure of personal information of victims was 8.9, that is, the degree of disclosure of personal information of perpetrators was higher than that of victims. According to the t-test results of the offender/victim personal information disclosure sample, there was a statistically significant difference between the offender and victim personal information disclosure scores. Overall, the offender's personal information was more open than the victim's. According to Tables 10 and 11, the results of the analysis of the information disclosure index showed that 'name' was the most disclosed personal information of the perpetrator. On the contrary, the "age" of the victim's personal information was the most exposed, which showed that the age of the victims was given more attention during the editing process.

Table 12 shows the related content of the persons in the sexual offences report that were analysed. According to Table 12, 34.8% of the sexual crime reports emphasised the offender and 15.7% emphasised the victim. When comparing the ratio of emphasis between the

**Table 9. Results of the disclosure of personal information of perpetrators/victims in news reports of sexual crimes by People's Daily Online.**

| Disclosure of personal information | Mean | Frequency | Standard deviation | Average standard deviation |
|---|---|---|---|---|
| Disclosure of personal information of perpetrators | 9.91 | 600 | 2.72 | .11 |
| Disclosure of personal information of victims | 8.93 | 600 | 2.10 | .09 |

Note: $t = 7.805$, $df = 599$, $p < .05$*

**Table 10. Analysis results of information disclosure index of perpetrators in sexual crimes news reports of People's Daily Online.**

| Disclosure of the perpetrator's personal information | Frequency | Mean | Standard deviation |
|---|---|---|---|
| (1) Name | 600 | 1.93 | .88 |
| (2) Age | 600 | 1.52 | .87 |
| (3) Face | 600 | 1.22 | .60 |
| (4) Address | 600 | 1.25 | .53 |
| (5) Employer | 600 | 1.64 | .90 |
| (6) Exposure to people around | 600 | 1.15 | .46 |
| (7) Exposure to private life | 600 | 1.21 | .54 |

Note: n = 600

**Table 11. Analysis results of information disclosure index of victims in sexual crimes news reports of People's Daily Online.**

| Disclosure of victim's personal information | Frequency | Mean | Standard deviation |
|---|---|---|---|
| (1) Name | 600 | 1.26 | .56 |
| (2) Age | 600 | 1.71 | .91 |
| (3) Face | 600 | 1.06 | .31 |
| (4) Address | 600 | 1.23 | .52 |
| (5) Employer | 600 | 1.47 | .83 |
| (6) Exposure to people around | 600 | 1.13 | .45 |
| (7) Exposure to private life | 600 | 1.07 | .30 |

Note: n = 600

**Table 12. Results of the analysis on the emphasis on relevant content of sexual crime reports in People's Daily Online.**

| Media attention | Frequency | Percentage (%) |
|---|---|---|
| 1) Neutrality | 297 | 49.5 |
| 2) Victim-related content emphasis | 94 | 15.7 |
| 3) Perpetrator-related content emphasis | 209 | 34.8 |
| Total | 600 | 100.0 |

Note: (unit: piece, (%))

perpetrator and the victim, the perpetrator was relatively more emphasised. Overall, however, neutral content accounted for 49.5%, or almost half, of sexual crime stories.

(7) Attribution of responsibility between perpetrators and victims in news reports related to sexual crimes

According to previous research [7], there are two kinds of analysis variables. First, the responsibility attribution of the perpetrator and the victim that has been specifically identified in the sexual crime reports. Second, the responsibility of the perpetrator or the victim has not been directly indicated. For example, "The female employee was drunk in the early morning and walked alone in the street in thin clothes". Although the direct cause of sexual crimes is not mentioned, one of the reasons for these reported sexual crimes is "a victim who is drunk and has thin clothes", which is a way to lure people into committing crimes, and the victim is partly responsible for the occurrence of the case. Based on the approach of antecedent research, the behavioral judgment of the victim or perpetrator of sexual crimes in context was followed, and who is to be blamed for the cause of the case was judged according to the attribution effect on a three-point scale, with 1 point for "not mentioned at all", 2 points for "mentioned to a certain extent" and 3 points for "specifically mentioned".

Tables 13 and 14 show the analysis results of the responsibility attribution of perpetrators and victims in the news reports of sexual crimes in People's Daily Online. First of all, 46.2% of the perpetrators were "not mentioned at all". On the contrary, 93.8% of the responsibility attribution to victims of sexual crime-related reports was "not mentioned at all". The overall results showed that more than half of the reported contents pointed to the responsibility of the perpetrators.

8) Thematic frame/episodic frame in news reports related to sexual crimes

The research on news frames focusing on news composition forms was divided into the thematic frame and episodic frame [27]. Based on the previous research, the focus of "What happened at what time and under what circumstances?" and "Who said what?" was on the report of a real list of cases, which was defined as the episodic frame of sexual crime news. Instead,

**Table 13. Analysis results of responsibility attribution to perpetrators in sexual crime news reports of People's Daily Online.**

| Attribution of responsibility to the perpetrator | Frequency | Percentage (%) |
|---|---|---|
| 1) Not mentioned at all | 277 | 46.2 |
| 2) Mentioned to a certain extent | 157 | 26.2 |
| 3) Specifically mentioned | 166 | 27.7 |
| Total | 600 | 100.0 |

Note: (unit: piece, (%))

**Table 14. Analysis results of responsibility attribution to victims in sexual crime news reports of People's Daily Online.**

| Attribution of responsibility to the victim | Frequency | Percentage (%) |
|---|---|---|
| 1) Not mentioned at all | 563 | 93.8 |
| 2) Mentioned to a certain extent | 31 | 5.2 |
| 3) Specifically mentioned | 6 | 1.0 |
| Total | 600 | 100.0 |

Note: (unit: piece, (%))

"Why did this happen?" "What are the main points of contention?" "What is the solution?", is the thematic frame that attempts to comprehensively analyze the causes, impacts, and solutions. Besides, "episodic frame + thematic frame" was added because of the double frame, and other frames were set as "Other".

According to Table 15, according to the results of the analysis on the frame frequency of the news reports on sexual crimes by People's Daily Online, 239(39.8%) articles adopted thematic frames, 315(52.2%) adopted episodic frames, and 42(7%) articles adopted both simultaneously. The episodic frame was more frequently used in sexual crime news.

## 5. Discussion and conclusion

This study's in-depth analysis of the coverage of sexual offences in the People's Daily Online between 2004 and 2019 reveals its multifaceted characteristics and trends. First, with a total of 13,897 stories recorded between 2004 and 2019, there was a notable shift in media coverage of sexual crimes. The period between 2004 and 2010 saw limited coverage. Only 71 stories dealt mainly with historical events such as comfort women and World War II incidents. The terminology used in People's Daily Online's coverage of sexual crimes varied according to the context of the Chinese language and legal jargon. Terms such as 'sexual assault', 'sexual crime' and 'rape' were used differently, reflecting either formal legal language or general narrative use. This is somewhat different from some of the reports from Japan and Korea, suggesting that there are cultural and legal nuances in the reporting of sexual crimes in some regions [31]. Since 2011, there has been a significant increase in number of reports. This increase suggests an increase in public awareness and a shift in media practices, consistent with the findings of Grøndahl et al. [32]. However, since 2016, there has been a noticeable decline in such coverage, possibly due to changes in media policy, public awareness and a focus on more nuanced reporting. This trend is observed by Christensen and Pollard [33] in their study of media representations of individuals involved in child sexual abuse material. The evolution in reporting

**Table 15. Results of the analysis on the frame of news reporting on sexual crimes by People's Daily Online.**

| Frames | Frequency | Percentage (%) |
|---|---|---|
| 1) Thematic frame | 239 | 39.8 |
| 2) Episodic frame | 315 | 52.5 |
| 3) Thematic frame + Episodic frame | 42 | 7.0 |
| 4) Other frame | 4 | .7 |
| Total | 600 | 100.0 |

Note: (unit: piece, (%))

also reflects changes in societal attitudes and media sensitivities, as demonstrated by Tufail's [34] study on the racialisation of sexual crime reporting.

In terms of regional coverage of sexual crimes and the reliability of sources, it was found that a significant proportion (60%) of stories came from regions outside mainland China. This suggests a global perspective in media coverage of sexual crime. As Krisdinanto et al. [35] illustrate, media coverage of sexual crimes is not limited to a single location or national perspectives, but extends globally.

At the same time, we find a pronounced reliance on government agencies as news sources in over half of the cases. This suggests a preference for authoritative voices, perceived as efforts to provide credible and trustworthy information. However, this trend also raises concerns about the lack of diversity of perspectives in reporting. As Bestari [36] points out, while the psychological impact of sexual violence and the government's role in victim recovery are crucial, the over-reliance on official sources may have limited the breadth of reporting and may not have adequately represented grassroots perspectives and victim-centred narratives.

The reporting style and celebrity involvement in People's Daily Online coverage of sex crimes, with a preference for traditional writing, showcases a conservative stance. However, this traditional approach might restrict the engagement and impact of these reports in today's visually oriented media landscape. Additionally, the focus of these reports is more on ordinary individuals than celebrities, perhaps to prevent the offence from becoming a widespread social sensation. This aligns with Zulfie et al.'s [37] study. While this approach is laudable for concentrating on broader societal issues over high-profile cases, it may also suggest a cultural or editorial bias in depicting crimes, potentially influencing public perception and dialogue.

When analysing whether the media focused on the perpetrator or the victim, we found that the media focused more on the perpetrator than on the victim. This suggests a serious imbalance in coverage, which could affect public perception and discussion of these issues. Baek and Yoon [38] also raise ethical considerations regarding the privacy of victims and the sensationalisation of crime, noting that while details of offenders are extensively revealed, information about victims is often vaguely covered. This approach risks fostering a culture of violence and voyeurism that could overshadow and surpass the dignified, empathetic and respectful coverage that victims deserve.

We conclude with an analysis of the framing of responsibility and the use of thematic/episodic frames. The analysis, which shows a higher attribution of responsibility to perpetrators than to victims, is consistent with contemporary discourses of accountability in sexual crime, as explored in Lindqvist's [39] research. This framing reflects evolving media narratives that emphasise perpetrator accountability and challenge historical patterns of victim blaming [40]. However, the use of both thematic and episodic frames in reporting suggests a nuanced approach to media representation, balancing individual stories with broader societal implications. As Grey and Killean [41] discuss, such framing strategies can have a significant impact on public understanding of sexual crime. Thematic framing can help contextualise these crimes within larger social and systemic issues, while episodic framing can provide a more intimate and immediate understanding of individual cases. However, the predominance of either frame can distort public perception, either by depersonalising the issue or by failing to address the systemic nature of the sexual crime.

Key areas for future research include an in-depth exploration of the ways in which sexual offences are reported and focused on in different geographical and cultural contexts, in order to provide a more comprehensive global perspective. There is also a need to analyse different news sources, including governments, non-specialist victims, social groups and public opinion, to ensure comprehensive coverage and multiple perspectives. At the same time, the project will focus on the impact of the People's Daily Online coverage of sexual crimes on public

perceptions and attitudes, particularly at a time when social media and digital technologies are becoming increasingly important. Exploring how to innovate reporting frameworks and methods to improve the quality of reporting and public engagement is also an important direction for future research. Finally, further research on journalistic ethics and responsibility, particularly in balancing truthfulness and comprehensiveness of reporting with respect and protection of victims, will be key to improving the quality of media coverage and social impact.

In conclusion, although the People's Daily Online has demonstrated multiple aspects in its coverage of sexual crimes, there is still room for improvement in terms of public impact, reporting strategies and journalistic ethics. In future media practice, there is a need to strengthen commitment to social responsibility and journalistic ethics to improve the quality of reporting and social impact, At the same time, progress can be made in achieving balance, depth and variety in coverage from a wider, more rigorous range of content.

## Author Contributions

**Conceptualization:** Yang Liu, Zhongzheng Fang.

**Data curation:** Yang Liu, Zhongzheng Fang.

**Formal analysis:** Zhongzheng Fang.

**Investigation:** Yang Liu.

**Methodology:** Yang Liu.

**Writing – original draft:** Yang Liu, Zhongzheng Fang.

**Writing – review & editing:** Yang Liu, Zhongzheng Fang.

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
