## [Decision Letter · Decision Letter 0]

9 Sep 2023

PONE-D-23-12920CHINESE SEXUAL VIOLENCE NEWS VIEWED FROM NETWORK NEWS: BASED ON NEW FRAME RESEARCHPLOS ONE

Dear Dr. Liu,

Thank you for submitting your manuscript to PLOS ONE. After careful consideration, we feel that it has merit but does not fully meet PLOS ONE’s publication criteria as it currently stands. Therefore, we invite you to submit a revised version of the manuscript that addresses the points raised during the review process.

This paper represents a documentary research focusing on cases of sexual violence reported in the People's Daily Online. Its topic is highly relevant and current, prompting deep reflection on the media's influence on society. Conversely, it is imperative to underscore that the material necessitates refinements, as highlighted by the two reviewers, particularly within the sections pertaining to methods, discussions, and conclusions.

We look forward to receiving your revised manuscript.

Kind regards,

Ricardo de Mattos Russo Rafael, Ph.D.

Academic Editor

PLOS ONE

Reviewers' comments:

Reviewer's Responses to Questions

**Comments to the Author**

1. Is the manuscript technically sound, and do the data support the conclusions?

Reviewer #1: Partly

Reviewer #2: Yes

2. Has the statistical analysis been performed appropriately and rigorously? 

Reviewer #1: Yes

Reviewer #2: Yes

3. Have the authors made all data underlying the findings in their manuscript fully available?

Reviewer #1: Yes

Reviewer #2: Yes

4. Is the manuscript presented in an intelligible fashion and written in standard English?

Reviewer #1: No

Reviewer #2: Yes

5. Review Comments to the Author

Reviewer #1: This is a documentary research on the news of sexual violence disclosed in the People's Daily Online. Its theme is current, relevant and brings reflections on how the media can influence society in a positive or negative way, depending on the content, format and the focus of the published reports.

However, the material lacks critical review and restructuring, especially with regard to the sections "Materials and Methods", "Dissucsion" and "Conclusion". subtitle formatting.

A detailed evaluation of the manuscript follows, with suggestions and recommendations to help authors improve the material.

Reviewer #2: The paragraph highlighted in yellow in the abstract needs to be reworked, because it was confusing to the reviewer. The first sentence in the introduction, “ … the actress …” is missing a name. Should the second last phrase, highlighted in yellow at the bottom of page 2, not read “social justice”? On page 7, the last sentence under (3), that reads “ … they will be highly concerned by the media” Is not clear to the reviewer. The author may want to revise this sentence or provide a clearer description for the reader to understand. The last sentence, highlighted in yellow, under “3. Research Questions” states: “This study will analyse …”. Was the study not already conducted? If yes, then it should state: “This study analysed …”. Can the author have a look at this?

6. PLOS authors have the option to publish the peer review history of their article (what does this mean?). If published, this will include your full peer review and any attached files.

Reviewer #1: No

Reviewer #2: No

---

## [Author Response · Author response to Decision Letter 0]

14 Nov 2023

We thank you and the reviewers for your review and valuable comments on our manuscript. During this revision process, we have structurally revised and supplemented the article to address each reviewer's comments. Please refer to reviewer#1 and reviewer#2 attachment. We have also reassessed the situation of the corresponding authors.

Hereby, we kindly request to change the corresponding author. After team discussion and consensus, we believe that changing the corresponding authors will better reflect the research work of this paper and ensure a smoother subsequent communication and revision process.

---

## [Decision Letter · Decision Letter 1]

26 Dec 2023

PONE-D-23-12920R1Reports on sexual violence published in an online Chinese newspaper: a new frame researchPLOS ONE

Dear Dr. Fang,

Thank you for submitting your manuscript to PLOS ONE. After careful consideration, we feel that it has merit but does not fully meet PLOS ONE’s publication criteria as it currently stands. Therefore, we invite you to submit a revised version of the manuscript that addresses the points raised during the review process.

We want to thank the authors for their diligent incorporation of the feedback provided by our ad hoc reviewers during the previous evaluation phase. The manuscript has demonstrated significant enhancements in terms of both structure and substance. Only a few minor adjustments are required to achieve its ultimate polish and refinement.

We look forward to receiving your revised manuscript.

Kind regards,

Ricardo de Mattos Russo Rafael, Ph.D.

Academic Editor

PLOS ONE

Journal Requirements:

Reviewers' comments:

Reviewer's Responses to Questions

**Comments to the Author**

1. If the authors have adequately addressed your comments raised in a previous round of review and you feel that this manuscript is now acceptable for publication, you may indicate that here to bypass the “Comments to the Author” section, enter your conflict of interest statement in the “Confidential to Editor” section, and submit your "Accept" recommendation.

Reviewer #1: (No Response)

Reviewer #2: All comments have been addressed

2. Is the manuscript technically sound, and do the data support the conclusions?

Reviewer #1: Partly

Reviewer #2: Yes

3. Has the statistical analysis been performed appropriately and rigorously? 

Reviewer #1: N/A

Reviewer #2: Yes

4. Have the authors made all data underlying the findings in their manuscript fully available?

Reviewer #1: Yes

Reviewer #2: Yes

5. Is the manuscript presented in an intelligible fashion and written in standard English?

Reviewer #1: Yes

Reviewer #2: Yes

6. Review Comments to the Author

Reviewer #1: This is a documentary research on the news of sexual violence disclosed in the People's Daily Online. Its theme is current, relevant and brings reflections on how the media can influence society in a positive or negative way, depending on the content, format and the focus of the published reports.

The authors met most of the suggestions and recommendations from the previous round of evaluation. However, the text of the Materials and Methods” and “Discussion and Conclusion” sections still requires minor revisions.

A detailed evaluation of the manuscript follows, with suggestions and recommendations to help authors improve the material.

Title and summary: The suggestions and recommendations of the previous opinion were met.

Introduction: The text of this section was completely restructured, as well as new sections were created: “2. Conceptual Definition and Related Literature”, which is a summary presented in topic format about the main concepts used in the study; 2.2 Related Literature, which is subdivided into other subtopics that present the knowledge already produced on topics relevant to the approach of the manuscript's object ("sexual crime reports”; "problems with sex crimes”; "obstacles to coverage”; and "the tool “frame” of news report analysis”).

- Specifically in relation to the subsection "2.1 Conceptual Definition", I emphasize that its content was displaced from the ideas discussed previously and in the following subsection. I recommend not using topics in this section and rewriting its content in plain text format, paying attention to the chain of ideas between paragraphs.

- Regarding section "2.2 Related Literature", I suggest not dividing it into subsections “(1) Literature review of sexual crime reports”, “(2) Problems with sex crimes” and “(3) Obstacles to coverage, presenting both contents as one text single.

- The wording of the objective presented in the introduction is not the same as that of the summary:

a)End of page 9: “The purpose of this study is to provide an in-depth analysis of the coverage of sexual crimes in recent years in China's main online media, the People's Daily”;

b)Abstract: "This study took People's Daily Online, a representative online media in China, as a research object to analyze the reporting of sexual crimes over the past 15 years”.

Materials and Methods: The text of this section was restructured and some recommendations from the previous opinion were met. However, I highlight that it still needs adjustments, as some important information was not presented.

- The text was reorganized into two subsections ("3.1 Research design”, "3.2 Material selection”), but the "Data analysis" section was missing, although its content is present in the text of subsection 3.2.

- In order to ensure the replicability of the study, I recommend that the authors make some clarifications about the second stage of the research process:

a) When did the data import phase happen? This phase was carried out by how many authors?

b) What was the instrument adopted to extract information from the reports? - By restructuring the text of this section, the authors suppressed the variables that guided the extraction of data from the selected news (Place of crime, the characteristics of sexual crime reports, The information source of a report, who is the persons in the news, the focus of media attention, Attribution of case responsibilities and frames focusing). who is the persons in the news, the focus of media attention, Attribution of case responsibilities and frames focusing).

Results: The text of this section was reorganized, maintaining the contents presented in the previous version of the manuscript.

Discussions and Conclusions: Despite the changes made to the text, it still does not reach the depth required for an analysis, as the authors propose.

- In the first three paragraphs, the authors summarize the findings of the study, already presented in the “Results” section.

- From the 4th to the 7th paragraphs, the authors make inferences about the findings in dialogue with the results of four studies. I recommend improving the discussion by incorporating other publications to discuss all the results, as some were not discussed.

- In the 8th paragraph, suggestions for new studies are presented.

- The 9th paragraph is dedicated to the conclusion.

References: Despite the addition of new references, I highlight that, of a total of 31 references, 23 are publications that are more than 5 years old. I recommend updating them.

Reviewer #2: All reviewer queries have been addressed adequately. Ensure no dual publication. The manuscript was presented in an intelligible fashion.

7. PLOS authors have the option to publish the peer review history of their article (what does this mean?). If published, this will include your full peer review and any attached files.

Reviewer #1: No

Reviewer #2: No

---

## [Author Response · Author response to Decision Letter 1]

27 Jan 2024

Dear Reviewer, 1

Reviewer #1: 

Introduction: The text of this section was completely restructured, as well as new sections were created: “2. Conceptual Definition and Related Literature”, which is a summary presented in topic format about the main concepts used in the study; 2.2 Related Literature, which is subdivided into other subtopics that present the knowledge already produced on topics relevant to the approach of the manuscript's object ("sexual crime reports”; "problems with sex crimes”; "obstacles to coverage”; and "the tool “frame” of news report analysis”).

- Specifically in relation to the subsection "2.1 Conceptual Definition", I emphasize that its content was displaced from the ideas discussed previously and in the following subsection. I recommend not using topics in this section and rewriting its content in plain text format, paying attention to the chain of ideas between paragraphs.

Reply: We thank you for your insightful comments on the structure and content of our manuscript, particularly in subsection '2.1 Concept Definition'. In response to your suggestions, we have made extensive changes to this section. Previously, the content was organized into topics which disturbed the flow and coherence of the ideas. To correct this, we have removed the topic-based formatting and reorganized the content into a continuous plain text format.

- Regarding section "2.2 Related Literature", I suggest not dividing it into subsections “(1) Literature review of sexual crime reports”, “(2) Problems with sex crimes” and “(3) Obstacles to coverage, presenting both contents as one text single.

- The wording of the objective presented in the introduction is not the same as that of the summary:

a)End of page 9: “The purpose of this study is to provide an in-depth analysis of the coverage of sexual crimes in recent years in China's main online media, the People's Daily”;

b)Abstract: "This study took People's Daily Online, a representative online media in China, as a research object to analyze the reporting of sexual crimes over the past 15 years”.

Reply: We thank you for your constructive feedback on the organization of section '2.2 Related Literature' and the consistency of the study's objective as presented in different parts of the manuscript. Following your recommendations：

1） We have removed the subdivisions within this section, namely '(1) Literature review of sexual crime reports', '(2) Problems with sex crimes', and '(3) Obstacles to coverage'. The content that was previously divided into these subsections has been seamlessly integrated into a single, cohesive text.

2） We have also addressed the discrepancy in the wording of the study's objective between the end of page 9 and the abstract. To ensure consistency, we have revised the wording so that both sections accurately reflect the same objective. We have standardized the term 'People's Daily Online' across the manuscript to avoid any ambiguity about the focus of our study.

Materials and Methods: The text of this section was restructured and some recommendations from the previous opinion were met. However, I highlight that it still needs adjustments, as some important information was not presented.

- The text was reorganized into two subsections ("3.1 Research design”, "3.2 Material selection”), but the "Data analysis" section was missing, although its content is present in the text of subsection 3.2.

- In order to ensure the replicability of the study, I recommend that the authors make some clarifications about the second stage of the research process:

a) When did the data import phase happen? This phase was carried out by how many authors?

b) What was the instrument adopted to extract information from the reports? - By restructuring the text of this section, the authors suppressed the variables that guided the extraction of data from the selected news (Place of crime, the characteristics of sexual crime reports, The information source of a report, who is the persons in the news, the focus of media attention, Attribution of case responsibilities and frames focusing). who is the persons in the news, the focus of media attention, Attribution of case responsibilities and frames focusing).

Reply: We appreciate your insightful feedback on the 'Materials and Methods' section of our manuscript. In response to your comments, we have made the following adjustments to provide greater clarity and ensure replicability of the study：

1） We acknowledge the omission of a distinct 'Data analysis' section. To rectify this, we have now added a new subsection titled '3.3 Data Analysis.'

2） We have included additional details about the data import phase. The text now states: 'The data collection, collation, cleanup, and import phase occurred from August 2021 to May 2022. All authors were actively involved in this process, ensuring a collaborative and comprehensive approach.‘

3） We have clarified the instrument used for data extraction. The revised text states: 'For extracting information from the reports, we employed the Scrapy framework, a powerful web-crawling tool written in Python. This framework allowed us to automate the extraction process efficiently. We defined specific data fields such as publication date, news content, and links using the Scrapy Item class. A specialized spider was programmed to search the People's Daily Online archive for terms related to sexual crimes. The extracted data was systematically processed and organized in Excel for further analysis.'

4） Recognizing the importance of detailing the variables that guided our data extraction process, we have reintegrated this information into the manuscript. We have also added Table 2, which succinctly presents these variables.

Results: The text of this section was reorganized, maintaining the contents presented in the previous version of the manuscript.

Reply: Thank you for your feedback regarding the 'Results' section of our manuscript. In response to your earlier comments, we have reorganized the text in this section to improve its clarity and coherence.

Discussions and Conclusions: Despite the changes made to the text, it still does not reach the depth required for an analysis, as the authors propose.

- In the first three paragraphs, the authors summarize the findings of the study, already presented in the “Results” section.

- From the 4th to the 7th paragraphs, the authors make inferences about the findings in dialogue with the results of four studies. I recommend improving the discussion by incorporating other publications to discuss all the results, as some were not discussed.

- In the 8th paragraph, suggestions for new studies are presented.

- The 9th paragraph is dedicated to the conclusion.

Reply: Thank you for your constructive feedback on the 'Discussion and Conclusions' section of our manuscript. We have taken your comments into account and have made the following changes to deepen our analyses and ensure a stronger discussion. We have reorganized the conclusions section to add a discussion of the results for all variables and to include additional references and perspectives that are consistent with our findings.

References: Despite the addition of new references, I highlight that, of a total of 31 references, 23 are publications that are more than 5 years old. I recommend updating them.

Reply: Thank you for your valuable feedback on the timeliness of the references in our manuscript. We recognize the importance of including current literature to enhance the relevance and current understanding of our area of research. In response to your suggestion, we have carefully reviewed our reference list and updated it accordingly. We have now included additional references from the last 5 years to ensure that our citations reflect the latest research and developments.

We believe that these revisions provide more thorough and detailed content to address the concerns you raised. We appreciate your guidance, which has been instrumental in improving the quality and depth of our manuscript.

Dear Reviewer, 2

Reviewer #2: All reviewer queries have been addressed adequately. Ensure no dual publication. The manuscript was presented in an intelligible fashion.

Reply: Your comments have been invaluable in guiding our revisions and ensuring the clarity and integrity of our work. 

Thank you again for your constructive feedback and guidance throughout the review process. We believe that your suggestions have significantly improved the quality of our manuscript.

---

## [Decision Letter · Decision Letter 2]

13 Feb 2024

Reports on sexual violence published in an online Chinese newspaper: a new frame research

PONE-D-23-12920R2

Dear Dr. Fang,

We’re pleased to inform you that your manuscript has been judged scientifically suitable for publication and will be formally accepted for publication once it meets all outstanding technical requirements.

Kind regards,

Ricardo de Mattos Russo Rafael, Ph.D.

Academic Editor

PLOS ONE

Reviewer's Responses to Questions

**Comments to the Author**

1. If the authors have adequately addressed your comments raised in a previous round of review and you feel that this manuscript is now acceptable for publication, you may indicate that here to bypass the “Comments to the Author” section, enter your conflict of interest statement in the “Confidential to Editor” section, and submit your "Accept" recommendation.

Reviewer #1: All comments have been addressed

2. Is the manuscript technically sound, and do the data support the conclusions?

Reviewer #1: Yes

3. Has the statistical analysis been performed appropriately and rigorously? 

Reviewer #1: Yes

4. Have the authors made all data underlying the findings in their manuscript fully available?

Reviewer #1: Yes

5. Is the manuscript presented in an intelligible fashion and written in standard English?

Reviewer #1: Yes

6. Review Comments to the Author

Reviewer #1: The authors followed all the suggestions and recommendations from the last round of evaluation. As a highlight, I recommend deleting the citation of the reference "Zhang, 2023” in the last paragraph, as it deals with the final considerations drawn up by the authors based on the findings.

7. PLOS authors have the option to publish the peer review history of their article (what does this mean?). If published, this will include your full peer review and any attached files.

Reviewer #1: No

---

## [Editor Report · Acceptance letter]

29 Apr 2024

PONE-D-23-12920R2 

PLOS ONE

Dear Dr. Fang, 

I'm pleased to inform you that your manuscript has been deemed suitable for publication in PLOS ONE. Congratulations! Your manuscript is now being handed over to our production team.

Kind regards, 

on behalf of

Dr. Ricardo de Mattos Russo Rafael 

Academic Editor

PLOS ONE